# *Ct*GH76, a Glycoside Hydrolase 76 from *Chaetomium thermophilum*, with Elongated Glycan-Binding Canyon

**DOI:** 10.3390/ijms26146589

**Published:** 2025-07-09

**Authors:** Silvana Ruth Ruppenthal, Wang Po-Hsun, Mohamed Watad, Christian Joshua Rosner, Marian Samuel Vogt, Markus Friedrich, Anna-Lisa Voigt, Angelique Petz, Petra Gnau, Lars-Oliver Essen

**Affiliations:** Faculty of Chemistry, Department of Biochemistry, Philipps-University, 35043 Marburg, Germany; silvana.ruppenthal@chemie.uni-marburg.de (S.R.R.); wangpo@staff.uni-marburg.de (W.P.-H.); watadm@staff.uni-marburg.de (M.W.); christian.rosner@chemie.uni-marburg.de (C.J.R.); marian_vogt@yahoo.de (M.S.V.); friedrich-regensburg@web.de (M.F.); anna-lisa.voigt@pharmazie.uni-marburg.de (A.-L.V.); angelique.petz@gmail.com (A.P.); gnaup@staff.uni-marburg.de (P.G.)

**Keywords:** glycoside hydrolases, glycan screening, cell wall, carbohydrate-binding proteins, protein families, biotechnology

## Abstract

Fungal cell walls, composed of polysaccharides and proteins, play critical roles in adaptation, cell division, and protection against environmental stress. Their polyglucan components are continuously remodeled by various types of glycosyl hydrolases (GHs) and transferases (GTs). In *Saccharomyces cerevisiae* and other ascomycetes, enzymes of the *Dfg5* subfamily, which belong as GTs to the GH76 family, cleave an α1,4 linkage between glucosamine and mannose to facilitate covalent linkage of GPI-anchored proteins to the cell wall’s polyglucans. In contrast, the functions of other fungal GH76 subfamilies are not understood. We characterized *Ct*GH76 from the sordariomycete *Chaetomium thermophilum*, a member of the *Fungi/Bacteria-mixed* GH76 subfamily, revealing conserved structural features and functional divergence within the GH76 family. Notably, our structural characterization by X-ray crystallography combined with glycan fragment screening indicated that *Ct*GH76 can recognize GPI-anchors like members of the *Dfg5* subfamily but shows a broader promiscuity toward other glycans with central α1,6-mannobiose motifs due to the presence of an elongated glycan-binding *canyon*. These findings provide new insights into GH76 enzyme diversity and fungal cell wall maturation.

## 1. Introduction

The fungal cell wall is a vital structural component and dynamic organelle that maintains cellular integrity, drives morphogenesis, and protects cells from environmental stress. Unlike mammalian cells, fungi have a unique cell wall, making them an attractive target for antifungal agents [1,2,3]. The biosynthesis of the fungal cell wall is a highly coordinated and complex process involving the synthesis and remodeling of core polysaccharide components, including chitin, glucans, and mannoproteins [1]. Glucans, primarily composed of β1,3-glucan with smaller amounts of β1,6-glucan, are the dominant polysaccharide constituents of the fungal cell wall [4,5]. These glucans are synthesized by transmembrane proteins that use nucleotide diphosphate sugars as substrates [6,7].

To ensure fungal cell wall functionality, various enzymes are critical for modeling and elongating polysaccharide precursors. These include GHs from diverse protein families, which catalyze the hydrolysis of glycosidic bonds in polysaccharides, facilitating the construction, modification, and degradation of cell wall components [2]. Their activities support cell wall maintenance, cell growth, and adaptation to mechanical and osmotic stress [1,8]. Alongside fungal polysaccharides, GPI-anchored cell wall proteins (GPI-CWP) are key components of the cell wall [9]. The common core structure of GPI anchors, found across protozoa, fungi, plants, and animals, comprises an ethanolamine phosphate (EtN-P), three mannose residues, a glucosamine, and an inositol phospholipid. Defects in GPI-anchor biosynthesis disrupt protein localization, impair cellular processes, and hinder the maturation of GPI-anchored proteins (GPI-APs), which involves endoplasmic reticulum (ER) translocation, GPI-anchor attachment, and post-translational modifications [10].

GHs are classified into over 189 families in the Carbohydrate-Active Enzymes (CAZy) database based on based on sequence similarity and structural characteristics [11]. However, detailed structural and functional analyses of GHs involved in cell wall maturation remain scarce. The GH72 family, prevalent in fungi, is crucial for yeast cell wall remodeling, acting as transglycosylases (GT) that elongate β (1–3) glucan chains. This process is regulated by a balance between glycosyl hydrolases, glycosyltransferases, and transglycosylases [12]. In *Neurospora crassa*, two of its nine GH72 proteins are implicated in cell-wall biosynthesis, cleaving and attaching N-linked outer-chain mannans to the cell wall [13,14]. Additionally, in *Bacteroides thetaiotaomicron*, α-mannan degradation involves GH92 (23 genes) and GH76 (10 genes) from the CAZy database. GH92 exo-α-mannosidases cleave α1,2 and α1,3 bonds, while GH76 enzymes specifically target the α1,6-mannan backbone, suggesting a model where GH92 removes branch-end mannosides and GH76 internally cleaves the backbone [11,15].

Recently, we demonstrated that the GH76 family functions extend beyond α1,6-mannanase activity. Members of the *Dfg5* subfamily act as GTs, facilitating the transfer of GPI-anchored proteins from the plasma membrane to β1,6-polyglucans by cleaving the Manα1,4-GlcN linkage in the GPI-core glycan [16,17]. These findings raised questions about the roles of other GH76 family members in hydrolysis and transglycosylation processes within the fungal cell. For example, the recently identified *Fungi/Bacteria*-mixed (*F/B*-mixed) subfamily [17], widely distributed among *Ascomycetes*, remains poorly understood. Here, we present the crystal structure of a novel fungal GH76 member from the *F/B*-mixed subfamily, in complexes with different glycans and glycan fragments. Together with structural bioinformatics, we found that members of the *F/B*-mixed subfamily act as secreted GHs in glycan degradation, which includes not only α1,6-mannans but also GPI-core glycans.

## 2. Results and Discussion

### 2.1. Sequence Similarity Network of the GH76 Family

The sequence similarity network (SSN) of the GH76 family, comprising 15,624 orthologs (Figure 1), is predominantly composed of fungal enzymes (9876, two-thirds) and bacterial representatives (5450, one-third). Within a GH76 subfamily, orthologs exhibit mostly pairwise sequence identities > 35%, a level of similarity sufficient to suggest a shared general function [18]. However, sequence identities between GH76 family members from distinct subfamilies can fall below 20%, entering the so-called “twilight zone” of protein sequence alignments, which further corroborates the notion of isofunctional subfamilies [19]. The largest GH76 cluster, the *Dfg5* subfamily (*Asco1*), includes 6301 sequences across 3190 nodes, almost exclusively from *Ascomycota*. This subfamily encompasses the structurally characterized *Ct*Dfg5 from *Chaetomium thermophilum* DSM 1495 (PDB: 6RY0). This enzyme is one of the six *Dfg5* paralogs in *C. thermophilum* and specifically recognizes GPI-core glycans in its active site, according to glycan fragment screening [17]. This type of fungal enzymes is vital as GT for ascomycete cell wall remodeling and processing of GPI-anchored cell wall-associated proteins (GPI-CWPs), indicating a critical role in fungal development and host interactions. Likewise, the bacterial subfamilies *Bacteria I* and *Bacteria II* are structurally characterized and exhibit validated α-1,6-mannanase activities as GHs. Accordingly, they are mostly involved in glycan degradation [20,21,22] (Figure 1). LamH from *Mycobacterium tuberculosis*, an ortholog of the GH76 subfamily *Bacteria III* (exclusive to Actinomycetes), catalyzes also the hydrolysis of α-1,6-mannosidic bonds in capsular lipomannan and lipoarabinomannans [23]. Here, this GH76 enzyme is suggested to play a role in growth phase transition due to a signaling function of released arabinomannans or fragments thereof.

The *Fungi/Bacteria*-mixed subfamily, the second-largest GH76 SSN cluster, comprises 2481 orthologs. Co-distribution analysis (see Venn diagram in Figure 1) reveals that 72.4% (761) of the 1051 ascomycete species possess at least one ortholog from both the *Dfg5* and *F/B*-mixed subfamilies. Apart from the *Dipodascomycetes* they all belong to the *Ascomycota* subphylum of *Pezizomycotina*, which contains filamentous ascomycetes like *Aspergillus* or *Neurospora* as well as most of the lichenizing fungi. Accordingly, the *Saccharomycetina* subphylum, encompassing most unicellular fungi (yeasts) [25], is absent from the *F/B*-mixed subfamily, which includes prominent classes such as *Saccharomycetes* and *Pichiomycetes*. The *Dfg5* and *F/B*-mixed subfamilies differ in paralog numbers per species, with *Dfg5* averaging 4.1 and *F/B*-mixed averaging 1.5 (Appendix A). Given the critical role of *Dfg5* members in cell wall biogenesis and stage-dependent maturation, the lower paralog counts in the *F/B*-mixed subfamily may suggest a broader function, such as glycan degradation. DeepTMHMM [26] and BIG-PI [27] analyses of ascomycete *F/B*-mixed subfamily members reveal a near-complete absence of transmembrane helices (1193/1253 orthologs) and ω-sites but the presence of an N-terminal signal peptide (1124/1253) for secretion. This contrasts sharply with the *Dfg5*-subfamily, whose members are anchored to the plasma membrane via a GPI anchor due to the presence of an ω-site or at least a transmembrane helix [17]. Overall, the predicted extracellular location of *F/B*-subfamily members is consistent with a function for glycan degradation rather than for cell wall maturation.

In the sordariomycete *C. thermophilum*, we found two *F/B*-mixed paralogs, *Ct*GH76 and *Ct*GH76B (Uniprot: G0S5Y9, G0S8A0). *Ct*GH76, with 439 amino acids, features an unusually long N-terminal signal peptide (M1-V60), as predicted by DeepTMHMM [26]. In contrast, *Ct*GH76B (389 aa) shares 35% sequence identity in its GH76 domain (N20-L384) but has only a short N-terminal signal peptide (M1-S18). Notably, *C. thermophilum* has another GH76 enzyme (Uniprot: G0SHT7), *Ct*GH76*, that belongs to the *Asco2* subfamily (Figure 1).

### 2.2. Overall Structure of CtGH76 and Its Active Site

Recombinant overproduction of *Ct*GH76, *Ct*GH76B, and *Ct*GH76* in *Escherichia coli* yielded soluble proteins, with highest yields for *Ct*GH76, that could be crystallized. *Ct*GH76 crystals belong to space group *P*3_1_21 with unit-cell parameters of *a* = 106.8, *b* = 106.8, *c* = 126.9 Å, *α* = *β*= 90°, and *γ* = 120° and one molecule per asymmetric symmetry unit. The *Ct*GH76 structure, determined by molecular replacement using the *Ct*Dfg5 structure as search model (PDB entry: 6RY0), displays the characteristic (α/α)_6_-helical barrel fold, featuring a core of six α-helices surrounded by another set of six α-helices. This structural feature is shared with other GH76 enzymes belonging to the fungal *Dfg5* or the bacterial GH76 subfamilies. Accordingly, *Ct*GH76 shares the same DD-motif, D163/D164 (Figure 2, black dashed box), in its active site as *Bc*GH76 from *Bacillus circulans* TN-31 (D124/D125) and *Ct*Dfg5 (D134/D135). For *Bt*3792, the GH76 enzyme from *Bacteroides thetaiotaomicron* VPI-5482^T^, these two neighboring aspartate residues were shown to participate in substrate hydrolysis via a retention mechanism [20]. Furthermore, the active site and the *canyon*-like glycan binding site of *Ct*GH76 is lined with seven aromatic amino acids (W110, W111, W149, Y217, W287, Y289, F352) that potentially interact with the substrate (Figure 2, red dashed box; for nomenclature, refer to Vogt et al. [17]), within an otherwise mostly negatively charged canyon [28] (Appendix A). *ConSurf* analysis of the *F/B-mixed* subfamily, mapped onto the *Ct*GH76 surface (Figure 2, grey dashed box), underscores the importance of the glycan-binding *canyon*, where nearly all residues show a high degree of conservation. The tertiary structure of *Ct*GH76 is stabilized by two disulfide bridges (Figure 2, blue dashed box), C204–C279 and C339–C345. C204 anchors the loop connecting helices α5 and α6 via its disulfide bridge to β-sheet β1, whereas the second disulfide bridge, C339–C345, is situated within the loop connecting helices α9 and α10.

*Ct*GH76 features two wing-like β-hairpins protruding between helices α7 and α8 (β1/2, D268-R283, designated as WR1) and between helices α11 and α12 (β3/4, R396-D413, WR2) (Figure 2). Only WR1 apparently elongates the glycan-binding canyon for providing additional sites of interactions with longer substrates. Otherwise, the *Ct*GH76 resembles other structurally known members of the GH76 family. Superposition with bacterial GH76 enzymes like the ortholog from *Bt*3792 (PDB: 4C1S) [20] and *Sh*GH76 from *Salegentibacter* sp. Hel_I_6 (PDB: 6SHD) [21] results in root-mean-square deviation (r.m.s.d.) values of 1.7 Å over 192 aligned Cα atoms and 1.3 Å over 255 aligned Cα atoms, respectively (Appendix A). While WR1 is present in these bacterial enzymes as well, other structures of the GH76 family lack both WR1 and WR2. Compared to the fungal *Ct*Dfg5 paralog, *Ct*GH76 exhibits an r.m.s.d. of 1.31 Å over 299 aligned Cα atoms and includes an additional short α-helix (α13) (Appendix A). Notably, the corresponding β-strands (i.e., β1 of *Ct*GH76 and β2 from *Ct*Dfg5) are tilted to each other. The presence of well-aligned structures with significant overlap raises questions about the specific functional role of *Ct*GH76. While *Bc*GH76 and *Sh*GH76 are characterized as typical α-mannanases, *Ct*Dfg5 acts as a transglycosidase for enabling the transfer of the GPI-core glycan to the non-reducing end of β1,3-glucans [17].

### 2.3. Glycan Fragment Mapping Reveals a Recognition Mode for GPI-Core Glycans

The mechanism by which fungal GH76 enzymes contribute to cell wall maturation was first elucidated through the characterization of *Ct*Dfg5. This enzyme of the *Dfg5* subfamily removes the GPI anchor of GPI-CWPs in a single step via a covalent intermediate, transferring the protein–glycan remnant to the non-reducing ends of the β1,3-glucans of the fungal cell wall [17]. An alternative hypothesis, based on the annotated α1,6-mannanase activity of bacterial homologs [22,29], proposes that some *Dfg5* enzymes cleave the α1,6-mannose backbone of N-linked outer chain mannans and transfer it to an acceptor glycan of the cell wall. For bacterial GH76 members, a minimum substrate length of three mannoses, i.e., a α1,6-mannotriose core, is required for hydrolysis, as seen with *Bc*GH76. Given that the precise function of *Ct*GH76 remains unclear, it raises the question of which glycan serves as the true substrate for this glycoside hydrolase from *Chaetomium thermophilum*. Our attempts to delineate a bacterial-like α1,6-mannasidase activity of *Ct*GH76 by a TLC assay using α1,6-mannotriose, -tetraose, and -hexaose as substrates failed. To investigate further for *Ct*GH76 specificity, we performed a glycan fragment screen against *Ct*GH76 following a protocol established before by us for *Ct*Dfg5 [17]. *Ct*GH76 crystals were soaked with GPI-glycan fragments (Figure 3A)—specifically mannose, α1,2-mannobiose, α1,6-mannotriose, and glucosamine. Based on the binding mode of the GPI-core glycan to *Ct*Dfg5, the ligand-binding site of *Ct*GH76 can be classified, spanning from the subsites +3 to −6 (Figure 3B; for nomenclature, refer to Vogt et al. [17]).

Soaking with mannose in *Ct*GH76 identified potential sugar-binding subsites analogous to *Ct*Dfg5, revealing the subsites −4, −2, and +3 (Figure 3B and Appendix A). Notably, soaking with α1,6-mannotriose yielded the same subsites (−4, −2, and +3), as they occupied three single mannose residues (Figure 3C and Appendix A). To exclude the residual α1,6-mannasidase activity of *Ct*GH76 under crystal soaking conditions, we generated single-site mutants for its catalytic DD motif. For the active-site mutants D164N (Figure 3D) and D163N (Figure 3E), soaking with α1,6-mannotriose resulted in α1,6-mannobiose binding to subsites −1/−2, alongside with single mannose residues at subsites +3 and −4 (Figure 3B). The presence of single mannose units in these subsites may result from either partial hydrolysis of mannotriose or impurities. When comparing the wild-type of *Ct*GH76 with its aspartate mutants after soaking with α1,6-mannotriose, sugar residues at subsites −4, −2, and +3 adopt nearly identical conformations. However, occupancy of subsite −1 is observed only in the active-site mutants, with slight variations in orientation (Appendix A). Furthermore, in *Ct*GH76 complexes with glucosamine and α1,2-mannobiose, glucosamine occupies subsite +1, interacting primarily with aromatic residues W110 and W111 and positioned above the general acid/base residue D164, while α1,2-mannobiose occupies subsites −2/−3 and an additional mannose is found at subsite −4 (Figure 3B and Appendix A). Overall, *Ct*GH76 appears to share a consistent binding mode across subsites −3 to +1, where the GPI-anchor core motif can be identified.

Although the residues interacting with the GPI-core glycan binding site are highly conserved between *Ct*GH76 and *Ct*Dfg5, several notable differences are evident: Firstly, the active site residue D164 in *Ct*GH76 is oriented outward compared to D135 in *Ct*Dfg5 (Figure 4A). As a result, D164 primarily interacts with the amino and 3-hydroxyl groups of glucosamine, whereas D135 forms strong hydrogen bonds with the 3- and 4-hydroxyls. Secondly, in *Ct*GH76, the 1- and 3-hydroxyls of glucosamine are associated with two glycerol molecules from the crystallization buffer, enhancing the structural integrity of the binding site. For the α1,6-mannobiose complex (Figure 4B), the subsite −2 displays a highly conserved binding mode for the mannose moiety, with interactions mediated by a conserved aspartate residue (*Ct*GH76: D268; *Ct*Dfg5: D250) that engages the 3- and 4-hydroxyls of mannose. Similarly, subsite −1 features stacking interactions between the mannose pyranose ring and a conserved aromatic residue (*Ct*GH76: W111; *Ct*Dfg5: W83). Interestingly, the mannose conformations at subsite −1 differ distinctly between *Ct*GH76 and *Ct*Dfg5 complexes. Firstly, *Ct*GH76 favors the β-anomer of the mannose 1-hydroxyl, whereas *Ct*Dfg5 prefers the α-anomer. This difference may result from the mutation of D164 in the DD motif to asparagine, causing its sidechain to swivel outward. Secondly, the mannosyl moiety in *Ct*Dfg5 adopts a regular ^4^C_1_ boat conformation, while in *Ct*GH76, the pyranose ring is distorted into an ^O^S_2_-like conformation. This strained conformation was previously observed in the bacterial GH76 ortholog Aman6 from *Bacillus circulans* in complex with α1,6-mannopentaose [22], where it was interpreted as being on the reaction pathway toward the B_2,5_ conformation of the GH transition state.

At subsites −2/−3 with bound α1,2-mannobiose (Figure 4C), aromatic residues surround the site to stabilize substrate binding. However, the mannose at subsite −3 in *Ct*GH76 adopts an inward orientation, likely due to two factors. Firstly, L284 in *Ct*GH76 replaces N262 of *Ct*Dfg5, and the lack of a hydrogen bond to the 3-hydroxyl may contribute to this shift. Secondly, at subsite −4, the terminal mannose’s O1 ether atom interacts with the 2-hydroxyl of the mannose at subsite −3, with a distance of 2.4 Å. For β1,3-glucobiose (Figure 4D), this β1,3-glucan fragment occupies subsites +2/+3 in *Ct*GH76, unlike the acceptor subsites +1/+2 in *Ct*Dfg5. In *Ct*Dfg5, the mannose at subsite +2 forms stacking interactions between its pyranose ring and the phenolic group of Y81 while being sandwiched between the α2 and α5 helices and enabling close proximity to the active site for GT activity via its covalent intermediate. In contrast, the absence of these α-helices in *Ct*GH76 reduces steric constraints, causing an outward shift of β1,3-glucobiose. Additionally, in *Ct*GH76, the mannose at subsite +2 forms stacking interactions with W149, while the 3- and 4-hydroxyls of the mannose at subsite +3 form hydrogen bonds with R103.

Computational simulations of the tetra-saccharide Manα1,2-Manα1,6-Manα1,4-GlcN, mimicking the GPI-core glycan, reveal multiple possible conformations, with key atom distances ranging from 5 Å, short r1’4-distances, to as long as 16 Å [30]. In *Ct*GH76, the reassembled GPI-core glycan adopts a C-shaped conformation, with a distance of 11.3 Å between the C1 atom of GlcN and the C4 atom of the third mannose, consistent with observations in *Ct*Dfg5 (Appendix A). In contrast, sugar substrates in *Bc*GH76 and *Sh*GH76 display linear and U-shaped conformations, with distances between the C1 atom of the first mannose and the C4 atom of last mannose measuring 19.4 Å and 7.8 Å, respectively (Appendix A). Compared to the fragment screen of *Ct*Dfg5, the current study confirms the ability of *Ct*GH76 to bind the same ligand at multiple positions within the binding canyon (Figure 3B and Appendix A). The primary difference in the comparison of GPI-core glycan poses lies in the slightly altered orientation of the glucosamine moiety and the increased bending in the third mannose (Appendix A).

The open binding site of *Ct*GH76 is consistent with the GPI-core glycan reassembly, suggesting that *Ct*GH76 may target GPI-core glycans as substrates, likely due to the abundance of these eukaryotic glycan fragments in the environment of the saprotrophic *Chaetomium thermophilum* species [31]. Glycan mapping revealed that the putative substrate binding site of wild-type *Ct*GH76 contains only single mannose units for α1,6-mannotriose, whereas DD motif mutants accommodate both mannose and α1,6-mannobiose within the binding canyon (Figure 3C–E). The α1,6-mannobiose at subsite −1 adopts an ^O^S_2_ conformation in *Ct*GH76, resembling a strained substrate, with a subtle shift likely caused by the active site mutation D164N (Appendix A). Reduced activities of DD motif mutants, as observed in corresponding *S. cerevisiae* Dfg5 mutants, result in a complete loss of function [17,22]. These findings indicate that *Ct*GH76 exhibits low substrate specificity, capable of hydrolyzing GPI-core glycans as a GH—distinct from the GT function of *Ct*Dfg5—and also acting as an α1,6-mannanase for α1,6-mannotriose and similar substrates. The presence of a signal peptide for secretion suggests that *Ct*GH76’s biological role and the specificity of *Chaetomium* is primarily linked to the degradation of glycans containing central α1,6-mannobiose or mannose-α1,4-glucosamine motifs.

### 2.4. Glycan Fragment Mapping Shows the Prolonged Glycan-Binding Canyon of CtGH76

The Dfg5-subfamily glycosyltransferases (GTs) are known to facilitate the transfer of GPI-anchored components from the plasma membrane, utilizing the GPI-core glycan as a donor, and cell wall glycans, such as β1,3-glucans or β1,6-glucans, as acceptors [32,33]. In *Ct*Dfg5, a binding site for Glcβ1,3-Glc (laminaribiose) was identified at the +1/+2 subsites, indicating a preference for β1,3-glucans as acceptors [17]. Notably, in *Ct*GH76, the disaccharide Glcβ1,3-Glc binds to two distinct sites (Figure 3B and Appendix A). First, it occupies the −1/−2 subsites in a manner like α1,6-mannobiose at the −2 subsite, but with a distinct orientation at the −1 subsite due to rotation of the glucosyl moiety at the reducing end. Surprisingly, Glcβ1,3-Glc also binds to the +2/+3 subsites, differing markedly from the +1/+2 subsite binding in *Ct*Dfg5 [17]. Overall, the glycan-binding canyon of *Ct*GH76 exhibits low specificity for glycans beyond α1,6-mannans and GPI-core glycans. This is supported by glycan mapping with monomeric glucose, which occupies the same binding sites as mannose (−4, −2, and +3; Appendix A vs. Appendix A), likely due to hydrophobic and electrostatic interactions between these subsites and the pyranose moieties. Similarly, α1,6-glucobiose (isomaltose) binds akin to Glcβ1,3-Glc and α1,6-mannobiose at the −1/−2 subsites near the active site’s DD motif (Figure 4E and Appendix A), indicating a rather high specificity of the subsites −1/−2 for an α1,6 glycosidic bond.

A distinctive feature of *Ct*GH76 is its extended β-wing region WR1, which is a candidate for additional glycan binding. Glycan fragment screening confirmed that α1,3-α1,6-mannotriose, a mimic of the core component of fungal high-mannose N-glycans, binds adjacent to WR1 at subsites −4/−5/−6 (Figure 5A). Specifically, its α1,3-mannobiose substructure at subsites −5/−6 interacts with WR1 residues A271 and Y280 (Figure 5B and Appendix A). The C6 carbon of the α1,6-linkage is stabilized by hydrophobic interactions with Y217, while L284 guides the orientation of the C2-C1-O1 atoms of the α1,3-linked mannose. Hydrogen-bonding interactions include T216 with the 2-hydroxy group of the α1,6-linked mannose and N282 from WR1 with the reducing end of the central mannose. Overall, WR1 provides additional subsites −5 and −6 otherwise missing in most GH76 orthologs, including members of the *Dfg5*-subfamily.

### 2.5. Comparison with Alphafold 3-Mediated CtGH76–Glycan Complex Models

Given the assembly-based approach of our glycan fragment screening method [17], it is valuable to investigate its synergy with de novo protein–ligand binding prediction using an all-atom model like AlphaFold 3 (AF3) [34]. When employing default settings, all five AlphaFold 3 models, each with six α1,6-mannobiose ligands bound to a CtGH76 molecule, consistently predict occupancy of the −1/−2, +1/+2, and −3/−4 subsites by mannobioses (14/30 ligands), while the remaining 16 mannobioses are distributed more heterogeneously across the protein surface. Notably, all models show the pyranose ring at subsite −1 adopting the ^4^C_1_ boat conformation, consistent with that observed in CtDfg5 (Figure 6A). In stoichiometric AF3 models of CtGH76 with α1,6-mannohexaose, four of five models predict a U-shaped arrangement spanning from subsite −3 to subsite +3, mirroring the U-shaped conformation of the GPI core glycan derived for CtDfg5 (Figure 6B). However, the fifth model, albeit occupying the same subsites, incorrectly predicts a reversed orientation in the glycan-binding canyon. Subsites −1/−2 exhibit the highest conformational homogeneity for α1,6-mannobiose across the models. In contrast, the U-shaped arrangement of α1,6-mannohexaose differs from the more linear distribution of the α1,6-mannobioses along the glycan-binding canyon in both AF3 models and experimentally determined subsites for glycan moieties in CtGH76. The validity of the latter, linear glycan conformations, is corroborated by a similar linear arrangement of the α1,6-mannopentaose in the glycan-binding canyon of Aman6 from B. circularis [22] (Figure 6C and Appendix A).

A limitation of this in silico approach is the potential bias by ligand poses present in AlphaFold 3’s training data [35]. This may account for the preference for U-shaped glycan conformations similar to known CtDfg5-binding poses, potentially limiting the ability of de novo predictions to identify additional binding sites, such as subsites −4 to −6, which are engaged by the α1,3-α1,6-mannotriose complex and primarily involve the WR1 motif. Notably, AlphaFold 3’s multimer mode, when used with peptide fragments, has shown greater success in modeling intrinsically disordered regions of protein ligands compared to full polypeptide ligands [36].

### 2.6. Structural Repertoire Within the F/B-Mixed Subfamily

AlphaFold3 [34] was also used to predict the structures of GH76 family members within the *F/B-*mixed subfamily, including orthologs from *Eutypa*, *Chaetomium*, *Ustilago*, *and Penicillium* species, representing diverse organisms such as *phytopathogens*, dead wood degraders, grass parasites, antibiotic producers, mold fungi, or simple model organisms [37,38]. Here, multiple sequence alignment of the *F/B*-mixed subfamily revealed that the N-terminus is less conserved than the C-terminus (Appendix A). This difference likely reflects distinct functional and structural roles within the enzyme. The N-terminus, often exposed and flexible, may facilitate diverse interactions with the cell wall or regulatory mechanisms, allowing for greater sequence variability. In contrast, the C-terminus, which typically contains the catalytic (α/α)_6_ toroid fold and the conserved DD motif, is essential for α-1,6-mannanase activity, thereby exhibiting higher sequence conservation.

The predicted structure of the *Eutypa lata* ortholog (Uniprot entry: M7SJ85) consistently exhibits the characteristic (α/α)_6_-helical barrel fold. Instead of a wing-like region, it features a lid-like structure composed of a long β-hairpin positioned over the glycan-binding canyon (Appendix A), a trait also observed in orthologs from *Xylaria* and *Chaetomium*. Such lid-like structures, seen in enzymes like lipases and Hsp70, enable transitions between closed and open states [39,40]. In lipases, for example, a closed lid shields the active site, restricting substrate access [40].

Structural predictions for members of the *F/B-*mixed subfamily from *Chaetomium globosum* reveal the conserved (α/α)_6_-barrel with an antiparallel β-hairpin motif on one side (WR1). However, the second region, WR2, present in some homologs, is absent in these orthologs. One *C. globosum* member (UniProt entry: Q2H301 [41]) displays a lid-like structure over the glycan-binding canyon and deviates from the typical GH76 fold, adopting a truncated (α/α)_5_-barrel instead (Appendix A). This ortholog also possesses a predicted ω-site, suggesting potential GPI-anchor modification (Appendix A). This conserved fold supports shared functionality within the *Chaetomium* genus.

In contrast, *F/B-*mixed subfamily members from the basidiomycete *Ustilago maydis* are predicted to be secreted likewise, with modeled structures exhibiting the characteristic (α/α)_6_-barrel of the GH76 family (UniProt entry: A0A0D1DNF6 [42]) (Appendix A), but lacking wing regions. For comparison, orthologs from *Penicillium brasilianum* are predicted to be membrane-associated by a GPI anchor due to the presence of a transmembrane helix and predicted ω-site (UniProt entry: A0A0F7VHI5 [43]) (Appendix A).

Overall, most analyzed structural models of the *F/B-*mixed subfamily predominantly share the (α/α)_6_-barrel fold, with one exception from *C. globosum* with its (α/α)_5_-barrel fold. Several proteins display a β-hairpin structure above the substrate-binding site. While the extended wing region observed in *Ct*GH76 is absent in most homologs, at least WR1 appears conserved within the *Chaetomium* genus. These findings suggest that members of the *F/B-*mixed subfamily may target similar complex glycan substrates.

## 3. Materials and Methods

### 3.1. Sequence Similarity Network Analysis of the CtGH76

This sequence similarity network illustrates the diversity and evolutionary relationships of the GH76 glycoside hydrolase family, based on the InterPro entry IPR005198. The network was constructed using the Enzyme Similarity Tool (EFI-EST) with an E-value threshold of 10^−40^, employing UniRef90 clustering. This resulted in a network corresponding to 8786 nodes with 15,624 sequences, connected by 5,296,761 edges, where each node represents sequences sharing at least 90% identity. The clusters highlight taxonomically and functionally distinct subfamilies, with several characterized proteins indicated.

### 3.2. Cloning, Overexpression, and Purification of CtGH76

The first 50 amino acids (M1-M50) were omitted. The codon-optimized gene, synthesized by BioCat GmbH (Germany), was cloned into the pET28a vector with an N-terminal His_10_-tag and a TEV cleavage site. Since the recombinant protein *Ct*GH76 contains disulfide bonds, expressing it in *E. coli* SHuffle T7 Express cells (New England Biolabs, Ipswich, MA, USA) can enhance soluble protein yield. Heterologous overexpression followed the protocol of Veelder et al. [44]. Cultures were grown in DYT medium at 37 °C to an OD_600_ of 0.3 and followed by cooling in ice water. Protein overproduction was induced with 10 µM IPTG at an OD_600_ of 0.5 and further incubated at 12 °C for 72 h. Harvested cells were resuspended in lysis buffer (20 mM HEPES, 300 mM NaCl, pH 7.5), flash-frozen in liquid nitrogen, and stored at −80 °C.

Cells were thawed, supplemented with PMSF and DNaseI, and disrupted using a French press (Aminco, Lake Forest, CA, USA). The centrifuged lysate’s sterile-filtered supernatant was loaded onto a 5 mL Ni-NTA column (GE Healthcare, Chicago, IL, USA). The column was washed with 5 column volumes (CV) of lysis buffer and 5 CV of washing buffer (lysis buffer + 25 mM imidazole). Protein was eluted with lysis buffer containing 250 mM imidazole. The His-tag was removed by overnight TEV protease digestion at 4 °C with rolling agitation. The digested *Ct*GH76 was reapplied to a 5 mL Ni-NTA column, and fractions containing the protein were pooled and concentrated using a 30 kDa cut-off Amicon Ultra concentrator (Millipore, Burlington, MA, USA). Final purification was achieved via size-exclusion chromatography on a Superdex 26/60 75pg column (GE Healthcare) with buffer (20 mM HEPES, pH 7.5, 100 mM NaCl) (Appendix A). Protein fractions were pooled, concentrated to 15 mg/mL (Amicon Ultra, 30 kDa cut-off), and stored at 4 °C for immediate use or flash-frozen in liquid nitrogen and stored at −80 °C. Purification steps were monitored by 12% SDS-PAGE (Appendix A).

Furthermore, the overproduction of all three GH76 orthologs from *C. thermophilum, Ct*GH76, *Ct*GH76B, and *Ct*GH76* was evaluated. Recombinant overproduction followed by Ni-NTA purification yielded less than 10 mg/L of culture for *Ct*GH76B and *Ct*GH76*, while *Ct*GH76 itself exhibited the highest purity and yield of ~30 mg/L of culture (Appendix A).

### 3.3. Crystallization of CtGH76 and Its Mutants

Crystallization experiments were conducted using the Honeybee 963™ robot (Digilab) with MRC 2-well sitting-drop plates (Swissci, Victoria, Australia). For each condition, 80 μL of precipitant solution was used as the reservoir, with 0.3 μL dispensed into the well, followed by 0.3 μL of protein solution at 15 mg/mL. Plates were sealed with foil and incubated at 20 °C in a Rock Imager (Formulatrix, Bedford, MA, USA). Wild-type *Ct*GH76 crystals formed in the JCSG Core IV E5 condition (3.6 M sodium formate, 10% (*v*/*v*) glycerol), appearing overnight or sooner (Appendix A). In contrast, *Ct*GH76 mutants D163N and D164N crystallized in the JCSG Core IV E8 condition (1.0 M sodium/potassium tartrate, 0.2 M lithium sulfate, 0.1 M Tris, pH 7.0) (Appendix A) but not in the wild-type condition.

### 3.4. Glycan Fragment Screening Using Crystal Soaking

Various glycans were dissolved in water at concentrations of 0.5 to 2 M, depending on solubility. The substrate was mixed 1:1 with the crystallization condition for the protein and its mutants before transferring single crystals to the soaking solution. Soaking duration was adjusted based on crystal behavior and halted upon observing dissolution or surface cracking (Appendix A). Crystals were harvested and flash-frozen in liquid nitrogen without cryoprotection.

### 3.5. Data Processing and Structure Determination

X-ray data were collected from single crystals at 100 K at beamline X06DA-PXIII (Swiss Light Source SLS, Villigen, Switzerland) and beamline ID23-1 and ID30-3 (European Synchrotron Radiation Facility ESRF, Grenoble, France). Datasets were processed using XDS [45]. Due to the polar space group *P*3_1_21, some datasets required reindexing using the *Reindex* program from the CCP4 suite [46]. The initial unbound *Ct*GH76 structure was solved by molecular replacement using Phaser [46], with *Ct*Dfg5 (PDB: 6RY0) [17] as the search model. Subsequent ligand-bound and mutant structures were determined by molecular substitution using the initial *Ct*GH76 structure as the starting model. Refinements, including isotropic *B*-factor and TLS refinement for the protein chain, were performed using phenix.refine from Phenix [47] and Coot [48]. Data collection, processing, and refinement details are provided in Appendix A. The resulting coordinates were analyzed and visualized using PyMOL [49].

### 3.6. Conservation Analysis

Conserved and variable amino acids were identified to predict functionally important protein regions. A multiple sequence alignment of the fungi/bacterial mixed subfamily was used for ConSurf analysis of the *Ct*GH76 structure [50]. The Bayesian method was selected for conservation scoring, and the neighbor joining with ML distance was used for phylogenetic analysis.

### 3.7. In Silico Prediction of CtGH76–Glycan Complexes Using AlphaFold 3

Structures of *Ct*GH76–glycan complexes were predicted using a local installation of AlphaFold3 [34]. The protein sequence (FASTA format) and glycans (SMILES code) were provided as input for de novo complex prediction. Default parameters were applied, using AlphaFold 3’s diffusion-based algorithm to generate high-confidence structures with predicted local distance difference test (pLDDT) scores being over 85. The resulting models were visualized and analyzed using PyMOL. Predictions were validated against experimental data to ensure accuracy.

## 4. Conclusions

In this study, we employed glycan fragment-mapping techniques to investigate the glycan-binding specificity of *Ct*GH76, shedding light on its potential role in fungal cell wall maturation. Compared to other GH76 orthologs, our findings demonstrate that *Ct*GH76 features an elongated glycan-binding *canyon*, adept at accommodating a wider range of complex glycans. This suggests that *Ct*GH76 may serve a dual role: functioning as an α1,6-mannanase on mannose-rich substrates while also potentially able to deal with GPI-core glycans, similar in its recognition mode to *Ct*Dfg5, but acting as a GH than a GT. The enzyme’s spacious and elongated glycan-binding *canyon*, capable of coordinating multiple ligands, highlights its remarkable adaptability to the diverse substrates commonly found in fungal environments. Nevertheless, it raises critical inquiries about substrate selectivity and cellular regulation—especially how *Ct*GH76 inhibits the premature degradation of functional GPI-anchored proteins in *Chaetomium* itself. Although this study sheds light on the structural foundation of *Ct*GH76’s glycan interactions, its precise physiological function remains uncertain, requiring further research into substrate preferences and regulation.

## Figures and Tables

**Figure 1 ijms-26-06589-f001:**
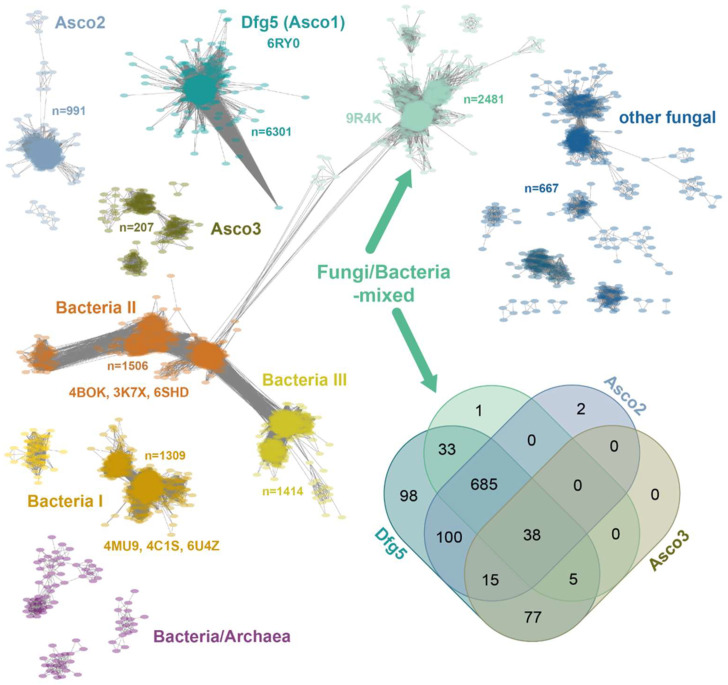
Sequence similarity network of the GH76 family (IPR005198, 15,624 sequences) depicting relationship among GH76 orthologs with an E-value cutoff of 10^−40^. Every node (colored oval) represents sequences of >90% identity, yielding 8396 nodes across nine major clusters and several smaller clusters. The Venn diagram illustrates the distribution of ascomycete species across four GH76 subfamilies, predominantly comprising ascomycete members [24]. The *Dfg5* subfamily (1051 organisms) is conserved across the Ascomycota due to its essential role in cell wall maturation [17].

**Figure 2 ijms-26-06589-f002:**
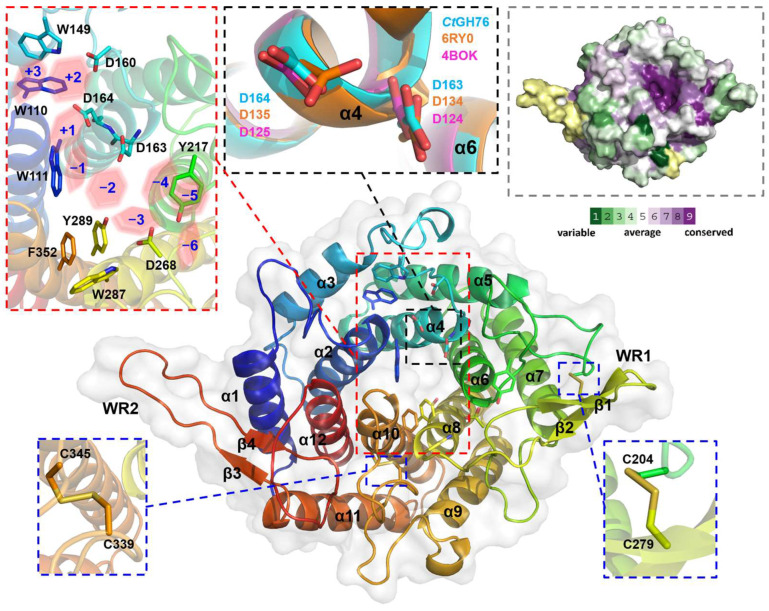
Crystal structure of *Ct*GH76 (apo form, PDB entry 9R4K) depicted as a cartoon model, colored in a rainbow gradient from N-terminus (blue) to C-terminus (red). Two extended β-hairpins, termed the β-wing region (WR), are highlighted, with secondary structure elements numbered. A close-up view of conserved aromatic amino acids in the active site is shown within a red-dashed box, and the subsites of the glycan-binding canyon are shown as red hexagons. The conserved catalytic DD motif is shown in a black-dashed box. The DD motif of *Ct*GH76 (cyan) is superimposed with *Ct*Dfg5 (PDB entry 6RY0, orange) and *Bc*GH76 (PDB entry 4BOK, magenta), represented as stick models. Conserved regions of *Ct*GH76 are shown in the grey-dashed box. The color legend applies to the surface representation, illustrating conserved and non-conserved regions.

**Figure 3 ijms-26-06589-f003:**
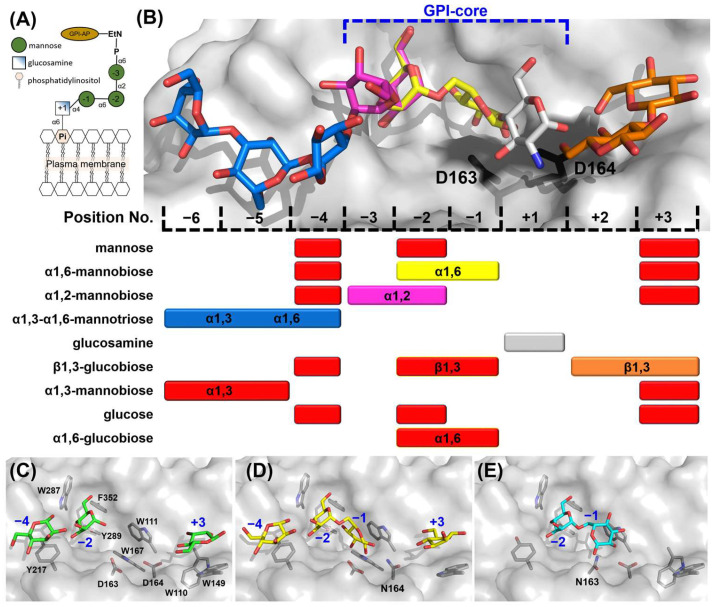
Mapping the GPI-core glycan using glycan fragment screening. (**A**) Schematic view of the GPI-anchor core structure; the depiction of the GPI-core glycan follows the Symbol Nomenclature for Glycans (SNFG). (**B**) Sugar residue positions in glycan-binding canyon are numbered, with glucosamine designated as position +1, identifying nine sugar-binding ligand positions in total. Colored mapping bars correspond to specific glycans: α1,6-mannobiose (yellow), α1,2-mannobiose (magenta), α1,3-α1,6-mannotriose (blue), glucosamine (grey), and β1,3-glucobiose (orange). Red bars indicate sugars not shown in the structural model above. The DD motif is depicted as black stick models in the active site. (**C**) Crystal structure of wild-type *Ct*GH76 (WT) soaked with α1,6-mannotriose (α1,6-Man), revealing three mannose residues (green stick models) occupying distinct subsites (PDB: 9R4M). (**D**) The D164N mutant soaked with α1,6-mannotriose shows intact α1,6-mannobiose (yellow) bound at the active site (PDB: 9R4P). (**E**) The D163N mutant soaked with α1,6-mannotriose exhibits a binding mode similar to the D164N mutant (PDB: 9R4O, cyan).

**Figure 4 ijms-26-06589-f004:**
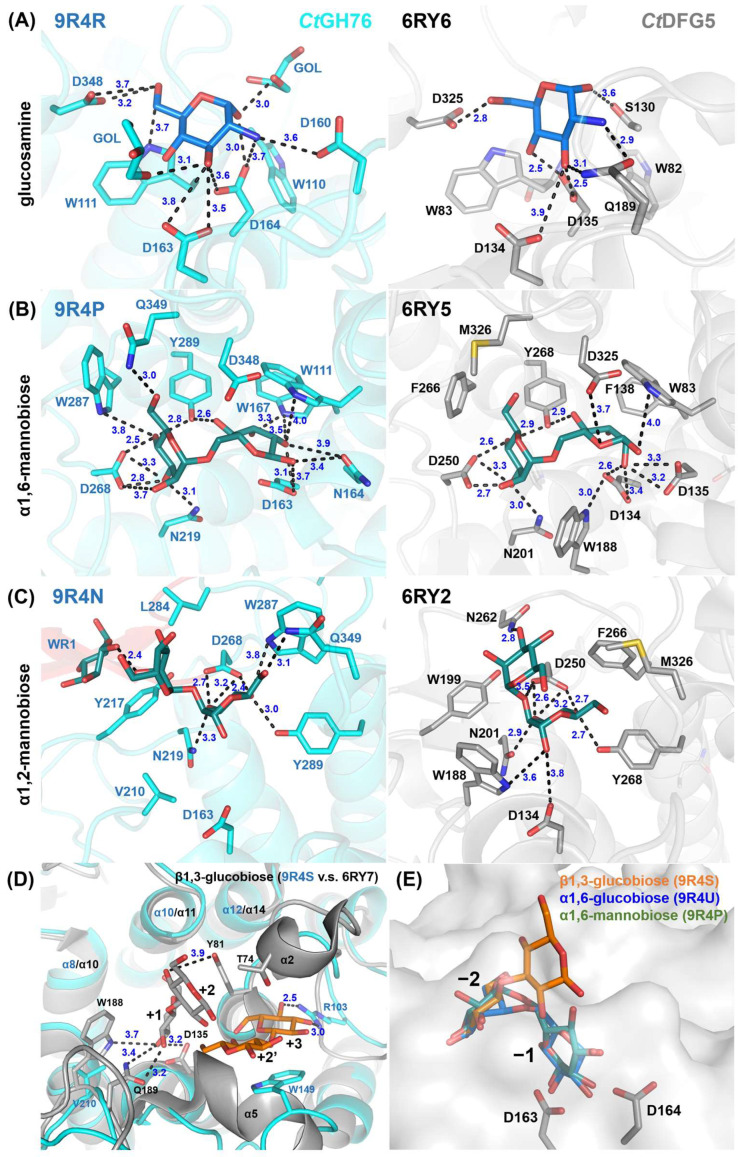
Detailed view of the glycan fragment mapping for the GPI-core glycan in *Ct*Dfg5 (grey) and *Ct*GH76 (cyan). Fragments of the GPI-core, including (**A**) glucosamine, (**B**) α1,6-mannobiose, and (**C**) α1,2-mannobiose, are depicted as stick models. Distances (in Å) between interacting residues and glycan fragments are indicated in black. (**D**) Superposition for β1,3-glucobiose in *Ct*Dfg5 (subsites +1/+2, grey) and in *Ct*GH76 (subsites +2/+3, orange). (**E**) Superposition for β1,3-glucobiose (orange), α1,6-glucobiose (green), and α1,6-mannobiose (blue) at subsites −1/−2. Notably, β1,3-glucobiose does not occupy properly the subsite −1 due to an outward orientation of the glucosyl moiety.

**Figure 5 ijms-26-06589-f005:**
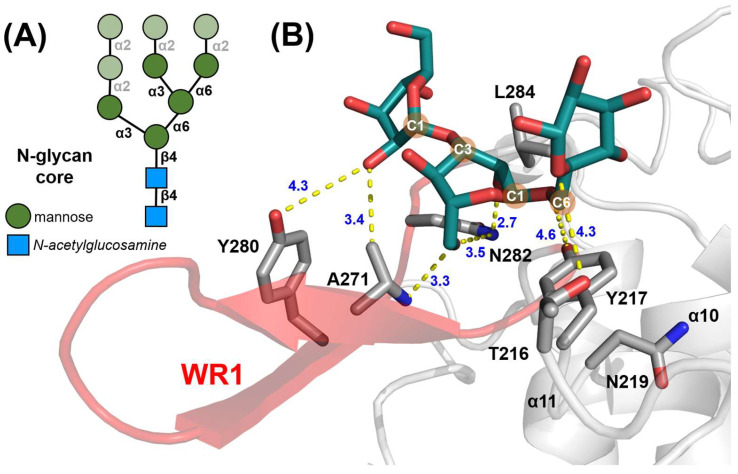
The β-wing region 1 (WR1) of *Ct*GH76. (**A**) Diagram of the high-mannose core of fungal N-glycans, with α1,3-α1,6-mannotriose substructures highlighted in dark green. SNFG symbols in the bottom left corner denote its building blocks. (**B**) Stick model of α1,3-α1,6-mannotriose (green) and its interacting residues within *Ct*GH76’s structure. The WR1 region is highlighted in red, with the remaining structure in grey.

**Figure 6 ijms-26-06589-f006:**
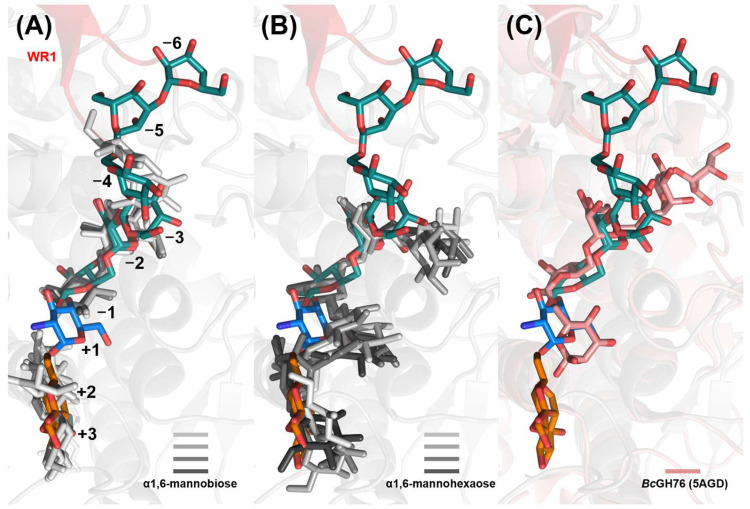
Comparison of predicted and observed *Ct*GH76–glycan complex models. Superposition of predicted stick models for (**A**) α1,6-mannobioses (light to heavy grey), (**B**) α1,6-mannohexaoses (light to heavy grey), and observed (**C**) α1,6-mannopentaose (5AGD, pink) with the observed *Ct*GH76–glycan complex (9R4R, blue; 9R4P and 9R4N, dark green; 9R4S, orange). The WR1 region is highlighted in red, with the remaining structure in grey.

## Data Availability

All data can be found in the Appendix A text. Protein structures have been uploaded to the PDB.

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
