# Peer review of "CtGH76, a Glycoside Hydrolase 76 from Chaetomium thermophilum, with Elongated Glycan-Binding Canyon"

_ijms, 2025, doi:10.3390/ijms26146589_

Round 1
Reviewer 1 Report
Comments and Suggestions for Authors
The article employs structural analysis to elucidate the interactions between CtGH76 and glycan, with robust experimental methods and well-substantiated results. I would like to discuss the following points with the authors:
The authors need to clarify what constitutes the "Fungi/Bacteria-mixed subfamily" in the Introduction section.
As the authors mentioned, the actual biological function of CtGH76 remains unclear. If possible, could you attempt to analyze CtGH76's activity toward GPI-core glycan substrates?
The paper mentions that the TLC assay was attempted but failed - could you explain why this occurred? This is typically a standard experimental method; alternatively, high-performance liquid chromatography could have been used as an alternative approach.
Why was low-temperature induction at 12°C employed during expression? The manuscript does not specify the conditions or timing for adding the inducer to the culture medium.
The experimental methods section lacks information about AlphaFold 3 modeling - this needs to be supplemented.
I cannot access the supplementary materials/figures.
Author Response
#reviewer1
The article employs structural analysis to elucidate the interactions between CtGH76 and glycan, with robust experimental methods and well-substantiated results. I would like to discuss the following points with the authors:
→Thank you for the review and comments, here are the point by point answers as following:
Q1: The authors need to clarify what constitutes the "Fungi/Bacteria-mixed subfamily" in the Introduction section.
A1: The referee is right. We included the Fungi/Bacteria-mixed subfamily description in the introduction section as following: “For example, the recently identified Fungi/Bacteria-mixed (F/B-mixed) subfamily [17], widely distributed among Ascomycetes, remains poorly understood.”
Q2: As the authors mentioned, the actual biological function of CtGH76 remains unclear. If possible, could you attempt to analyze CtGH76's activity toward GPI-core glycan substrates?
A2: The referee is right. However, we would like to point out that the assaying against GPI-core glycans is non-trivial, as these glycans are commercially not available and their supply as synthetic glycans remained very scarce (Tsai et at. 2011; doi: 10.1002/anie.201103483). Furthermore, the mass-spectrometric approach, used by us in our work on Dfg5 (ref.17, Vogt et al., PNAS 2020) is non-quantitative. Currently, to assess comparatively the activity of fungal GH76 family members on GPI-core glycan substrates, we are setting up an in vitro assay using S. cerevisiae GPI-anchored proteins. However, this cannot be the scope of the current manuscript on CtGH76.
Q3: The paper mentions that the TLC assay was attempted but failed - could you explain why this occurred? This is typically a standard experimental method; alternatively, high-performance liquid chromatography could have been used as an alternative approach.
A3: The referee is right. We attempted to perform TLC experiments by using α1,6-mannotriose, –tetraose and –hexaose but failed. We cannot exclude that these ligands may be incorrect substrates for CtGH76. To assess the in vitro activity of CtGH76, indeed, we conducted HPLC-based assays using only α-1,6-mannobiose as the substrate which we clearly characterized in the structure. However, no mannose signal was detected, indicating again a lack of hydrolysis by using α1,6-mannobiose. Thompson et al. (2015, DOI:10.1107/S1399004714026443) reported similarly that mannobiose is not a substrate for BcGH76, with enzymatic activity increasing with longer mannose chains. This suggests that only longer glycans are subject to enzyme recognition and effective hydrolysis.
Q4: Why was low-temperature induction at 12°C employed during expression? The manuscript does not specify the conditions or timing for adding the inducer to the culture medium.
A4: To enhance the expression of the disulfide-bonded protein CtGH76, we used E. coli SHuffle T7 Express cell (DOI: 10.1186/1475-2859-11-56) for protein overexpression. The unique properties of SHuffle T7 cells enabled expression at lower temperature, promoting slower expression and improved protein folding. We have revised the material and method section to reflect these details as following: “Since the recombinant protein CtGH76 contains disulfide-bonds, expressing it in E. coli SHuffle T7 Express cells (New England Biolabs) can enhance soluble protein yield. Heterologous overexpression followed the protocol of Veelder et al. [44]. Cultures were grown in DYT medium at 37°C to an OD600 of 0.3 and followed by cooling in ice water. Protein overproduction was induced with 10 µM IPTG at an OD600 of 0.5 and further incubated at 12°C for 72 hours.”
Q5: The experimental methods section lacks information about AlphaFold 3 modeling - this needs to be supplemented.
A5: We have added up AlphaFold3 modeling information at Method section 3.7.
Q6: I cannot access the supplementary materials/figures.
A6: We are very sorry from the missing files uploading in the beginning, it should have been uploaded by the editor a couple of days ago.
Reviewer 2 Report
Comments and Suggestions for Authors
The manuscript titled "CtGH76, a glycoside hydrolase 76 from Chaetomium thermophilum with an elongated glycan binding canyon" presents a structural characterization achieved through X-ray crystallography and glycan fragment screening. The results indicate that CtGH76 can recognize GPI-anchors similarly to members of the Dfg5 subfamily, while also exhibiting a broader affinity for other glycans featuring central α1,6-mannobiose motifs, attributed to its elongated glycan binding canyon. These discoveries offer valuable insights into the diversity of GH76 enzymes and the maturation of fungal cell walls.
In summary, the study demonstrates a high standard of research quality. Through comprehensive structural analysis, it explores the glycan-binding specificity of CtGH76, illuminating its potential role in the maturation of fungal cell walls. I find this research very interesting, as these structural analyses will also offer valuable insights for the discovery of specific inhibitors.
Author Response
#reviewer2
The manuscript titled "CtGH76, a glycoside hydrolase 76 from Chaetomium thermophilum with an elongated glycan binding canyon" presents a structural characterization achieved through X-ray crystallography and glycan fragment screening. The results indicate that CtGH76 can recognize GPI-anchors similarly to members of the Dfg5 subfamily, while also exhibiting a broader affinity for other glycans featuring central α1,6-mannobiose motifs, attributed to its elongated glycan binding canyon. These discoveries offer valuable insights into the diversity of GH76 enzymes and the maturation of fungal cell walls.
In summary, the study demonstrates a high standard of research quality. Through comprehensive structural analysis, it explores the glycan-binding specificity of CtGH76, illuminating its potential role in the maturation of fungal cell walls. I find this research very interesting, as these structural analyses will also offer valuable insights for the discovery of specific inhibitors.
→Thank you for your insightful feedback on the manuscript, we greatly appreciate your review of the research quality and its contributions to understanding GH76 enzymes and fungal cell wall maturation. Your comment on the potential for discovering specific inhibitors is particularly encouraging for us, and we agree that these structural insights could guide future inhibitor development. Thank you again for your review.
Reviewer 3 Report
Comments and Suggestions for Authors
This study investigates the structural and functional features of CtGH76, a glycosyl hydrolase from Chaetomium thermophilum, within the context of the diverse GH76 enzyme family. The authors present a clear comparison between CtGH76 and the better-studied Dfg5 subfamily in S. cerevisiae, highlighting a key divergence in substrate recognition. Using X-ray crystallography and glycan fragment screening, the study demonstrates that CtGH76 retains the ability to bind GPI-anchor–related motifs but shows broader specificity due to a uniquely extended glycan-binding groove.
The work is well-aligned with ongoing efforts to understand fungal cell wall biogenesis and glycosylation diversity. It offers valuable structural insights into the underexplored Fungi/Bacteria-mixed GH76 subfamily and contributes to the broader understanding of fungal cell wall remodeling. The novelty lies in identifying the binding promiscuity of CtGH76 and its potential role beyond classical GPI-anchor processing.
Overall, this is a well-executed and meaningful study with clear structural and functional relevance. However, the manuscript would benefit from additional biochemical validation of CtGH76’s enzymatic activity and potential physiological roles in C. thermophilum.
I have few minor comments listed below.
- In line, 140 authors discussed different yields by constructs. Please mention the yields in mg/lit in material method section.
- In line 144, please add the PDB ID for CtDfg5 structure.
- Please label the residue positions in the figure S2A.
- IN Figure2, please correct the subsite labels as they are tilted and difficult to visualize.
- Line 225, is it mannose or mannobiose, please correct it.
- Line, 240 it looks like in figure S3F, the substrate occupies -4 subsite also.
- Figure 4, keep the bond colors black and consistent for all the figures for better visualization.
- Figure 4 keep the ligand color unique and same in both CtDfg5 and Ct GH76.
- Line 405, the authors analyzed the sequences based on similarity network, which is informational, but showing the sequence alignment of different proteins of the family will give clear representation of the conserved residues and differences in the sequences. The differences and conservation can be explained using predicted structures now.
- There is no information about the affinity of the ligands how tight they bind to the active site. Whether is it already done in other studies if not it would be very insightful to do invtro binding studies, like ITC.
Author Response
#reviewer3
This study investigates the structural and functional features of CtGH76, a glycosyl hydrolase from Chaetomium thermophilum, within the context of the diverse GH76 enzyme family. The authors present a clear comparison between CtGH76 and the better-studied Dfg5 subfamily in S. cerevisiae, highlighting a key divergence in substrate recognition. Using X-ray crystallography and glycan fragment screening, the study demonstrates that CtGH76 retains the ability to bind GPI-anchor–related motifs but shows broader specificity due to a uniquely extended glycan-binding groove.
The work is well-aligned with ongoing efforts to understand fungal cell wall biogenesis and glycosylation diversity. It offers valuable structural insights into the underexplored Fungi/Bacteria-mixed GH76 subfamily and contributes to the broader understanding of fungal cell wall remodeling. The novelty lies in identifying the binding promiscuity of CtGH76 and its potential role beyond classical GPI-anchor processing.
Overall, this is a well-executed and meaningful study with clear structural and functional relevance. However, the manuscript would benefit from additional biochemical validation of CtGH76’s enzymatic activity and potential physiological roles in C. thermophilum.
I have few minor comments listed below.
→Thank you for the review and comments, here are the point by point answers as following:
Q7: In line, 140 authors discussed different yields by constructs. Please mention the yields in mg/lit in material method section.
A7: Referee is right. We made an additional supplementary figure (See Fig. S8) to illustrate different relative yields during the purification of these orthologs. Also, we mentioned the yields in the material and method section as “Furthermore, the overproduction of all three GH76 orthologs from C. thermophilum, CtGH76, CtGH76B and CtGH76*, was evaluated. Recombinant overproduction followed by Ni-NTA purification yielded less than 10 mg/L of culture for CtGH76B and CtGH76*, while CtGH76 itself exhibited the highest purity and yield of ~30 mg/L of culture (Fig. S8).”
Q8: In line 144, please add the PDB ID for CtDfg5 structure.
A8: Thank you, it’s done.
Q9: Please label the residue positions in the figure S2A.
A9: Thank you. The residues within the active site are now labeled. Please see figure S2A.
Q10: In Figure2, please correct the subsite labels as they are tilted and difficult to visualize.
A10: Thank you, it’s done. The subsite labels are non-tilted now.
Q11: Line 225, is it mannose or mannobiose, please correct it.
A11: Thank you, it is the mannose soaking experiment. It is now written as “Soaking with mannose in CtGH76…..”
Q12: Line, 240 it looks like in figure S3F, the substrate occupies -4 subsite also.
A12: Referee is right. We revised the sentence to “while α1,2-mannobiose occupies subsites -2/-3 and an additional mannose is found at subsite -4”.
Q13: Figure 4, keep the bond colors black and consistent for all the figures for better visualization.
A13: The referee is right. It’s done. They are color consistent in both now.
Q14: Figure 4 keep the ligand color unique and same in both CtDfg5 and CtGH76.
A14: The referee is right. It’s done. They are color consistent in both now.
Q15: Line 405, the authors analyzed the sequences based on similarity network, which is informational, but showing the sequence alignment of different proteins of the family will give clear representation of the conserved residues and differences in the sequences. The differences and conservation can be explained using predicted structures now.
A15: The referee is right. We added a sequence alignment for these proteins (See Fig. S5). We wrote a section to supplement as following: “Here, multiple sequence alignment of the F/B-mixed subfamily revealed that the N-terminus is less conserved than the C-terminus (Fig. S5). This difference likely reflects distinct functional and structural roles within the enzyme. The N-terminus, often exposed and flexible, may facilitate diverse interactions with the cell wall or regulatory mechanisms, allowing for greater sequence variability. In contrast, the C-terminus, which typically contains the catalytic (α/α)\₆ toroid fold and the conserved DD motif, is essential for α-1,6-mannanase activity, thereby exhibiting higher sequence conservation.
Q16: There is no information about the affinity of the ligands how tight they bind to the active site. Whether is it already done in other studies if not it would be very insightful to do in vitro binding studies, like ITC.
A16: The referee is right. However, we have not conducted binding assays for CtGH76 with the glycan fragments, as these are expected to exhibit only low affinities in the high mM to M range (for example refer to Veelders et al., PNAS 107:22511-22516, 2008). Other issues are (1) the often entropic nature of glycan binding prohibiting ITC analysis and (2) the multiple subsites, to which glycan fragments were found to bind to. With the availability of a long substrate-like glycan these shortcomings can be addressed, but this is a subject of further work.